# Amyotrophic Lateral Sclerosis and Air Pollutants in the Province of Ferrara, Northern Italy: An Ecological Study

**DOI:** 10.3390/ijerph20085591

**Published:** 2023-04-20

**Authors:** Annibale Antonioni, Vittorio Govoni, Lisa Brancaleoni, Alessandro Donà, Enrico Granieri, Mauro Bergamini, Renato Gerdol, Maura Pugliatti

**Affiliations:** 1Unit of Clinical Neurology, Department of Neurosciences and Rehabilitation, University of Ferrara, 44121 Ferrara, Italy; annibale.antonioni@edu.unife.it (A.A.); enrico.granieri@unife.it (E.G.); 2Doctoral Program in Translational Neurosciences and Neurotechnologies, University of Ferrara, 44121 Ferrara, Italy; 3Department of Environmental and Prevention Sciences, University of Ferrara, 44121 Ferrara, Italy; 4Preventive Medicine and Risk Assessment, University of Ferrara, 44121 Ferrara, Italy

**Keywords:** amyotrophic lateral sclerosis (ALS), pathophysiology, environmental factors, air pollutants, heavy metals, copper, neurodegeneration

## Abstract

The etiopathogenesis of amyotrophic lateral sclerosis (ALS) is still largely unknown, but likely depends on gene–environment interactions. Among the putative sources of environmental exposure are air pollutants and especially heavy metals. We aimed to investigate the relationship between ALS density and the concentration of air pollution heavy metals in Ferrara, northern Italy. An ecological study was designed to correlate the map of ALS distribution and that of air pollutants. All ALS cases diagnosed between 2000 and 2017 (Ferrara University Hospital administrative data) were plotted by residency in 100 sub-areas, and grouped in 4 sectors: urban, rural, northwestern and along the motorway. The concentrations of silver, aluminium, cadmium, chrome, copper, iron, manganese, lead, and selenium in moss and lichens were measured and monitored in 2006 and 2011. Based on 62 ALS patients, a strong and direct correlation of ALS density was observed only with copper concentrations in all sectors and in both sexes (Pearson coefficient (*ρ*) = 0.758; *p* = 0.000002). The correlation was higher in the urban sector (*ρ* = 0.767; *p* = 0.000128), in women for the overall population (*ρ* = 0.782, *p* = 0.000028) and in the urban (*ρ* = 0.872, *p* = 0.000047) population, and for the older cohort of diagnosed patients (2000–2009) the assessment correlated with the first assessment of air pollutants in 2006 (*ρ* = 0.724, *p* = 0.008). Our data is, in part, consistent with a hypothesis linking copper pollution to ALS.

## 1. Introduction

Amyotrophic lateral sclerosis (ALS) is a neurodegenerative disease characterised by a progressive drop in motor neurons in the cerebral motor cortex, brain stem and anterior horns of the spinal cord, leading to a gradual loss of voluntary motility [1,2]. The progression of this deficit causes fatigue, fasciculations, dysphagia, a loss of autonomy in daily activities and, finally, respiratory failure and death [3]. There are several variants of ALS, which differ according to the nature (upper and lower motor neuron) and localisation of the motor neurons primarily involved at the onset [4,5]. Multiple factors can affect the disease phenotype, such as age and gender [6]. Many hypotheses have been formulated about the mechanisms involved in motor neuron death, such as oxidative stress, glutamatergic excitotoxicity, mitochondrial dysfunction, a deficient elimination of toxic products and abnormal protein accumulation [7,8,9]. Currently, ALS is considered a complex multifactorial disease, caused by the interaction between genetic factors, in particular, SOD1, C9orf72, TARDBP and FUS, among the many known ones, and environmental factors [10,11,12]. A number of case–control studies have shown possible—yet controversial—associations with a large number of risk factors such as a rural vs. urban area of residence [13,14], electromagnetic fields [15], occupation (e.g., welders and farmers) [16,17], toxins from heavy metals, chemicals, solvents and herbicides used in agriculture [17,18,19]. Among the most commonly studied environmental factors is the exposure to heavy metals, such as Copper (Cu), Magnesium (Mg), Aluminium (Al), Manganese (Mn), Mercury (Hg), Lead (Pb), Selenium (Se), Cadmium (Cd), and Iron (Fe), for their possible role as cofactors in the pathogenesis of ALS due to numerous mechanisms, such as their interactions with enzymatic and metabolic activities and/or the neurotoxicity resulting from their excessive accumulation [20,21,22]. Indeed, transitional metals with redox activity feature many biological processes, as they are cofactors for a number of enzymes, including superoxidodismutatsis (SOD).

The aim of our study was to correlate the distribution of prevalent ALS cases in the municipality of Ferrara with the concentration of air pollutants in the same territory in a comparable time interval, and by sex and time.

## 2. Materials and Methods

### 2.1. Study Design

An ecological, or correlational, type of study was used to explore the correlation between aggregates of exposure and the disease. The study design implies the identification of geographical maps within which aggregates of exposure and disease concentration can be correlated in comparable time intervals.

### 2.2. Study Population

The study was conducted in the municipality of Ferrara, a town in the northwest of Italy with a population of 132,000. The climate is temperate and sub-continental, with an average annual temperature of 13.2 °C and average precipitation of 650 mm. Winds are predominantly from the west. In the Ferrara health district (HD), the estimated mean annual ALS incidence was almost stable over the period 1964–2009 at a rate of 1.82 per 100,000 inhabitants [23].

### 2.3. Ascertainment of ALS Cases

ALS cases were ascertained by using administrative data from the Ferrara University Hospital’s regional health information system. ICD-9M code 352.20 was used for hospital admissions, including access to the day hospital and day services for the period 2000–2017 were. Personal code, sex, place and date of birth were then obtained, as well as place of residence upon diagnosis, with post code. Diagnoses through ICD-9M code 352.20 were then validated against clinical records by neurologists M.P. and V.G. The exclusion criteria were (i) misclassified diagnosis under ICD-9M code 352.20, (ii) a lack of information on the place of residence upon diagnosis, and (iii) a place of residence upon diagnosis that was out of the study territory. Given the official administrative nature of the source of data, the patients’ residence addresses were up to date at the time of the study.

### 2.4. Measurement of Air Pollutants

The air pollutants of interest for this work were measured in previous investigations conducted by Gerdol et al. in the municipality of Ferrara [24]. In particular, biomonitoring techniques for these elements were based on the use of passive living organisms (due to their different sensitivity to polluted air) and/or active ones (due to their ability to accumulate pollutants in their tissues) [25,26]. The organisms used were mosses, vascular plants and lichens. Mosses, in particular, have the ability to accumulate pollutants, especially metals [25]. In the study area, Gerdol et al. found the presence of 23 different elements within mosses: Aluminium (Al), Silver (Ag), Arsenic (As), Barium (Ba), Bismuth (Bi), Cadmium (Cd), Cobalt (Co), Chrome (Cr), Copper (Cu), Iron (Fe), Gallium (Ga), Lithium (Li), Manganese (Mn), Molybdenum (Mo), Nickel (Ni), Lead (Pb), Rubidium (Rb), Antimony (Sb), Selenium (Se), Strontium (Sr), Uranium (U), Vanadium (V), and Zinc (Zn). The distribution of these elements in the study area varied according to the distance from the emission foci (in particular the petrochemical pole) and in relation to the direction of the wind [27]. For the moss samples, Gerdol et al. used only one species, *Tortura Muralis* Hedw, an acrocarpic moss frequently found in the seas and rocky soils of northern Italy. Why this type of moss is considered suitable for this kind of study and how it is measured is reported elsewhere in detail [28,29,30]. An estimate of air quality was based on the lichen diversity value (LDV). The LDV was determined using the method officially recognised by the Italian Agency for Environmental Protection and Technical Services [31,32].

### 2.5. Correlation Matrices and Study Maps

Ferrara is located in the Emilia-Romagna Region, in the northwest of Italy, one of the most densely populated regions in Europe. Pollutant sources in the urban and suburban area are mainly located in an industrial area that hosts several factories, especially those of chemicals. This area is located to the northwest of the urban area. However, pollutants are also regularly released from other sources, such as garages, painting facilities, domestic heating, and traffic, and both on urban roads and the nearby motorway. Moreover, considering the many farms, in this area, pollutants are released from crops due to intensive agricultural practice, such as through the use of agricultural machinery or widely dispersed pesticides and fertilizers. The concentration of pollutants in the air was within the legal limits, according to measurements carried out by the local environmental company in 2011. The sample analysis was conducted in a 10 × 10 km area, in accordance with the guidelines of ANPA (2001) [31]. The total area was then divided into 100 sub-areas of 1 sqkm. When possible, mosses and lichens were sampled in 250 × 250 m areas throughout the various sub-areas [24] (Figure 1).

In order to investigate the correlation between ALS and environmental air pollutants, the sampling site used by Gerdol et al. was divided into four sectors. The squares were grouped into the following four sectors, based on the distribution of CORINE land cover classes [33] (Figure 2):

The urban sector (exposed to downwind from the west winds), comprising 24 areas: 15/1, 6/2, 14/1, 14/2, 13/4, 13/3, 13/1, 13/2, 8/2, 8/3, 8/4, 12/1, 9/4, 9/1, 2/4, 12/2, 9/3, 9/2, 2/3, 11/1, 10/4, 10/1, 11/3, and 11/2;The highway sector (which ran from southwest to northwest of the urban sector), comprising 12 areas: 16/1, 16/2, 17/1, 17/2, 18/1, 18/2, 18/3, 19/4, 19/3, 20/4, 20/3, and 21/2;The northwest sector (which has some crops, industrial sites, a landfill site and a photovoltaic site, all exposed to south-easterly winds), comprising 19 areas: 25/4, 25/1, 16/4, 15/3, 25/3, 25/2, 16/3, 15/2, 24/4, 24/1, 17/4, 14/4, 24/3, 24/2, 17/3, 14/3, 23/4, 23/1, and 18/4;The rural sector (exposed to downwind from the west winds), comprising 42 areas: 5/4, 5/1, 6/1, 5/3, 5/2, 7/4, 7/1, 4/4, 4/1, 7/3, 7/2, 4/3, 4/2, 8/1, 3/4, 3/1, 3/3, 3/2, 2/1, 2/2, 1/4, 1/1, 1/2, 1/3, 10/2, 10/3, 19/1, 12/4, 19/2, 12/3, 20/1, 11/4, 20/2, 23/3, 23/2, 22/4, 22/1, 22/3, 22/2, 21/4, 21/1, and 21/3.

This area was carefully studied in 2005 to identify the most suitable sites for specimen collection. While mosses were abundant in all areas, few sites were suitable for sampling lichens. The biomonitoring data refer to two samplings: one conducted in the period September–October 2006 and one in the period September–October 2011.

### 2.6. Correlation Analyses between ALS Case Distribution and Air Pollutants

Correlation analyses between ALS and air pollutant concentrations [24] were conducted for areas with an adequate number of ALS cases. The number of cases diagnosed with ALS for the period 2000–2017 (17 years) was mapped for each sub-area belonging to a given sector, and was summed up for each sector (Figure 3).

Case density was obtained by dividing the number of cases for each sector and the reference population of that sector (e.g., the estimated average annual population for the urban sector multiplied by the 17-year observation period) by 100,000 inhabitants. The same estimate was obtained separately for the male and female population, and also separately by a cohort of ALS diagnoses, based on the median of the year of diagnosis (year 2010)., i.e., for the period 2000–2009, and for the period 2010–2017. With regard to air pollutants, of the 23 monitored, only Cu, Al, Pb, Se, Cd, Fe, Ag, Cr and Mn were considered for analysis, since they are potentially involved in neurodegeneration [34]. Of these, the point values measured in 2006 and 2011 were considered, as well as the average value of the two periods. The correlation between the density of ALS cases and the concentration of pollutants was carried out by means of a bivariate correlation analysis with the concentration of pollutants in 2006 and 2011 and the mean, and also by stratifying by gender, by sector and by period of ALS diagnosis. For each relationship, the correlation coefficient (ρ) was calculated with a two-tailed test (*p* = 0.05) and its square, the expression of variance. The statistical analysis was carried out using the IBM SPSS Statistics programme, version 20 [35].

## 3. Results

### Descriptive Statistics

For the period 2000–2017 and through ICD-9M code 352.20 (motor neuron disease), 96 (51 men and 45 women, M:F = 1.13) cases of ALS diagnosed in the municipality of Ferrara were available for the study. After validation against medical records, 13 (13.5%) subjects were excluded because they were affected by another disease, such as multiple sclerosis, unspecified amyotrophy, spinal arthropathy, visual disturbance, respiratory insufficiency, lingual carcinoma, poliomyelitis, severe scoliosis, benign fasciculations and diffuse cramps. Moreover, 21 subjects were excluded because they were residents in areas outside the municipal area of interest, or had no valid address (Figure 4).

The analysis was conducted on 62 subjects, 32 (51.6%) men and 30 (48.4%) women, with a mean (SD) age at diagnosis of 70.4 (11.6) and 70.3 (12.3) years, respectively (the *p*-value was non-significant). Of the 62 considered subjects, 38 (61.3%) and 24 (38.7%) were diagnosed in the period 2000–2007, and 2008–2017, respectively. Due to the insufficient sampling of cases in the northwest (*N* = 1) and in the motorway (*N* = 3), a correlation analysis was performed for the urban (*N* = 54) and rural (*N* = 13) sectors, respectively (Table 1).

The results of the correlation analysis between ALS density and air pollutants, in relation to the total study area (*N* = 62), to the urban and rural sectors, and to gender and period of diagnosis are shown in Table 2 (Table 2).

A direct and statistically significant correlation was observed only with copper (Cu) for the whole population, particularly for the urban sectors, the oldest diagnosis cohort and the 2006 copper measurements (Table 2). The correlation coefficients for the general population ranged from 0.712 (rural sector) to 0.767 (urban sector), indicating a variance (*ρ*^2^) of 51–59% in the correlation between ALS and average copper concentrations from 2006–2011. Using the 2006 copper concentrations, the data gave a coefficient of 0.758 (*p* = 0.000002) for the total number of cases, 0.608 for the number of men affected by ALS (*p* = 0.002) and 0.782 for the number of women affected by ALS (*p* = 0.000028). No significant correlation was obtained using the 2011 copper concentration data. A similar trend was observed for the correlations in the urban sector (Table 2), with ^a^ correlation coefficient equal to 0.767 (*p* = 0.000128) for the total population, 0.587 (*p* = 0.017) for men and 0.872 (*p* = 0.000047) for women in the 2006 copper measurements. These data explain the variance of 63% for the total population, 47% for the male population with ALS and up to 73% for the female population. No significant correlation was observed with the 2011 copper measurements. Very weak correlation data with copper were obtained for the total rural population (coefficient = 0.712, *p* = 0.048), but were not stratified by sex, nor were the measurements in 2006 and 2011.

The concentrations of the heavy metals included in the analyses, in relation to year (2011 vs. 2011) and area (rural vs. urban), are shown in Table 3. A two-way ANOVA showed statistically significant differences in terms of years only for Fe and Mn, while for Se, statistically significant differences were found both by year and by area, and by the interaction between these two factors.

## 4. Discussion

A number of heavy metals are currently known in the literature to be involved in the pathogenesis of ALS, including those investigated in this study. Our study has not documented significant data on the role of Pb, Hg, Al, Mn, Se, Zn, Fe and Mg in the pathophysiology of motor neuron disease, and this confirms the great variability in results between the available studies [36,37,38,39,40]. Our work was aimed at studying a correlation between atmospheric pollutants and ALS, using an ecological study design; ALS patients diagnosed between 2000 and 2017, provided by clinically validated health flows, were considered in the context of the territorial mapping of the municipality of Ferrara, carried out to detect the presence and variability of pollutants over time.

Our data suggest an interaction between copper concentrations and ALS density in the municipality of Ferrara, especially for the urban sector and for older inhabitants (Cu measurements from 2006 and ALS diagnoses from 2000–2009). These data support the biomolecular evidence of the role of copper in neurodegeneration. Copper is a highly reactive transition metal and it is essential for metabolism, as it is present in the active sites of numerous enzymes (e.g., SOD1, cytochrome C oxidase, and ceruloplasmin) [41]. Its presence in free-form or in excess can catalyse various reactions, such as Haber–Weiss and Fenton ones, resulting in oxidative stress and cellular damage [42,43], and its incorrect distribution has equally negative effects [44]. The brain, in particular, is extremely susceptible to oxidative stress, and alterations in copper homeostasis are often associated with neurodegenerative processes, as evidenced by numerous pathologies affecting the central and peripheral nervous system (e.g., familial ALS and Alzheimer’s disease) [45]. Initially, the involvement of copper was only linked to its high oxidoreductive potential, which, associated with SOD1 mutations, could induce oxidative stress and explain, at least in part, the toxic action of mutated proteins [46]. Accordingly, copper chelators were used in the treatment of transgenic mice expressing mutated SOD1 associated with familial ALS, leading to an increase in animal viability and an improvement in general condition [47,48,49]. However, other studies investigating the effects of a decrease in copper levels in animal models using triethylenetetramine tetrahydrochloride (C6H22Cl4N4), a specific copper chelator, or by mutations in ATP7A (essential in copper metabolism), showed that the disease’s onset was indeed delayed, but that the decrease in copper levels was not able to effectively counteract the disease, and the animals still died [50,51]. The role of altered copper homeostasis in this disease, however, is still poorly understood. Some ALS patients have reduced copper levels in serum and cerebrospinal fluid [52], while others are characterised by low systemic copper levels [53]. Other researchers, however, report an accumulation of metals, including copper, during the disease’s progression [54].

Moreover, excitotoxicity, another key pathological mechanism involved in the onset of ALS, results from the reduced uptake of glutamate by astrocytes, with its consequent accumulation in the intercellular microenvironment [55]. This phenomenon could also be linked to copper homeostasis at the synaptic level; altered copper levels could cause deregulation of NMDA receptors, as copper acts as their high-affinity, voltage-dependent antagonist [56]. The excitotoxic stimulus induces in motor neurons the production of oxygen and nitrogen free radicals, leading to a level of oxidative stress that is harmful to the cell [57]. Moreover, evidence of the downregulation of the glial glutamatergic transporter EAAT2 in disease-affected areas, both in human sporadic ALS and in transgenic mice for mutated SOD1, led to the hypothesis of its role in ALS progression [58,59].

The SOD1 gene is located on chromosome 21, encodes a 16–18 kDa subunit that binds Cu^2+^ and a Zn^2+^ and, as associating with homodimers, forms the active form of the enzyme. When SOD1 is synthesised by the cell, it does not contain copper and zinc, which will only be acquired at a later stage. The experiments showed that the normal human SOD1 gene, in the absence of the two metal ions, forms large, stable protein oligomers under physiological conditions. The researchers also found that intermolecular disulphide bridges are formed during oligomerisation, implicating binding betwee Cys-6 and Cys-111 [60]. Thus, it is possible that the key event in ALS pathogenesis is the failure to bind Cu^2+^ and Zn^2+^, for both mutated and normal forms of SOD1, which is followed by the formation of homodimers that lead to motor neuron death [61]. Indeed, mutation affects a region (the beta barrel plug) that is critical for the structural stability of the protein. A solution study of the mutant enzyme revealed the increased plasticity of the apoprotein—i.e., the copper- and zinc-free polypeptide—possibly responsible for its increased tendency to aggregate in concentrated solutions [62]. These studies further confirm that it is the metal-free forms—perhaps attributable to an excess of Cu or Zn, also considering the antagonism between these two metals, since an excess of copper hinders the assimilation of zinc and vice versa—of SOD1 that cause the toxic aggregations in motor neurons [63].

Our study is in line, at least partially, with the descriptive study conducted by Govoni et al. in the territory of the health district of Ferrara on the distribution of incident cases of ALS in the period 1964–2009 by area of residence at the time of ALS onset, type of ALS and gender. Similarly to the work of Govoni et al., in fact, in which no gender differences were found in the urban population, our study also does not show gender differences in the correlation with copper concentrations, which were, however, much more significant for the urban population [23]. The differences highlighted by this study in relation to the area of residence require further investigation, as highlighted by the previous work of Govoni et al. (a survey on the number of ALS cases in relation to the level of urbanisation and habitual occupation) [13].

ALS incidence in the province of Ferrara is consistent with that in other European countries, i.e., 1–3/100,000 pop./year [23,64,65]. Santurtún et al. showed, in a Spanish population, a clear positive relationship between ALS and air Pb levels [65], pointing to more than one air heavy metal pollutant being associated with increased ALS risk [66,67]. Stable ALS mortality and hence incidence rates over the past decades have also been reported overseas. A recent study in New Zealand showed a mortality rate of 2.8/100,000 pop./year in the period 1992–2013 with 74% of the New Zealand population being made up of European and Caucasian descendants [68]. In the US population, very similar incidence and mortality rates are reported with the exception of Midwest where registered mortality from ALS is significantly higher [69,70,71].

Notably, the pathogenesis of ALS is multifactorial and depends on a complex (and only partially understood) combination of genetic, environmental and metabolic factors [19,72,73,74,75]. It is therefore possible to speculate that Cu acts synergistically with other risk factors, and that only after exceeding a certain critical threshold does it contribute to the risk of ALS onset [75]. Moreover, the biomonitoring data considered refer to two samplings: one conducted in the period September–October 2006 and one in the period September–October 2011. In this time interval, there were important changes in the air, which may have significantly altered atmospheric pollutant emissions. In particular, in November 2007 and February 2008, two new incinerator lines were activated. Moreover, in October 2010, a combined cycle energy plant was also activated. It is possible to hypothesise that changes in emissions due to the opening of these new facilities may have contributed, at least in part, to the non-significance of the data collected in the second time interval.

A limitation of our study is the lack of data on the work activity of the patients, which meant we could only hypothesise the possible relation between metals and ALS pathogenesis on the basis of the highly significant correlation observed, which suggests an interaction. Furthermore, our study has all the limitations of any ecological study, because it is based on group (and not individual) analyses, thus only allowing hypotheses to be generated. Misclassification in groups could have led to confounding factors or co-linearity [76]. In addition, this type of study is very vulnerable to ecological phallacy because inferences cannot be made between individuals and groups [77]. Moreover, although ecological study designs are more suitable for common and widespread diseases, several examples of such a methodology used to investigate rare diseases and diseases with multifactorial aetiology are reported in the literature, e.g., [78,79,80]. In our study, we privileged correlating ALS concentrations with outdoor rather than indoor pollutants given their potential higher impact on the disease’s pathogenesis, as shown elsewhere [81,82,83]. Finally, ALS is a relatively rare disease and the cases considered are consequently small in number [84]. As far as exposure factors are concerned, it should be emphasised that pre-existing data were used, excluding Hg from the survey, which is now considered an important element in the literature [85].

The strengths of our work can be attributed to the accuracy and completeness of the demographic data about ALS patients living in this area; furthermore, no a priori hypotheses were formulated, but they were generated at the population level, as the study was not motivated by post hoc observations between copper concentration and ALS cases. Furthermore, this was a multidisciplinary study, involving neuroscience, epidemiology and environmental science. We performed measurements in two time periods following Gerdol et al.’s model [24,28] and used the descriptive epidemiological data produced by Govoni et al. [13,23].

## 5. Conclusions

In conclusion, our data suggest that, in highly exposed areas, the presence of heavy metals, such as copper, could influence the susceptibility to ALS occurrence in the general population. However, in order to confirm this hypothesis, further interdisciplinary studies are needed to extend the correlation analysis to the entire Emilia-Romagna region or to investigate the association between ALS and copper at an individual level. According to the above-mentioned hypotheses, it will be possible to design studies that include wider territories or that are based on analyses of individuals, also considering work activities, clinical phenotypes of the disease, and copper concentrations in tissues and biological fluids. Furthermore, the results of this study, if confirmed, could also explain at least in part, the potential role of heavy metals in the aetiopathogenesis of ALS. Lastly, it would be interesting to analyse the possible role of metals emitted by new facilities opened in the period between the two measurements, in order to understand the possible ‘protective’ capacity (as copper-chelants or other) of certain elements against copper and thus explain the relevance of copper in 2006, which was instead absent in 2011.

## Figures and Tables

**Figure 1 ijerph-20-05591-f001:**
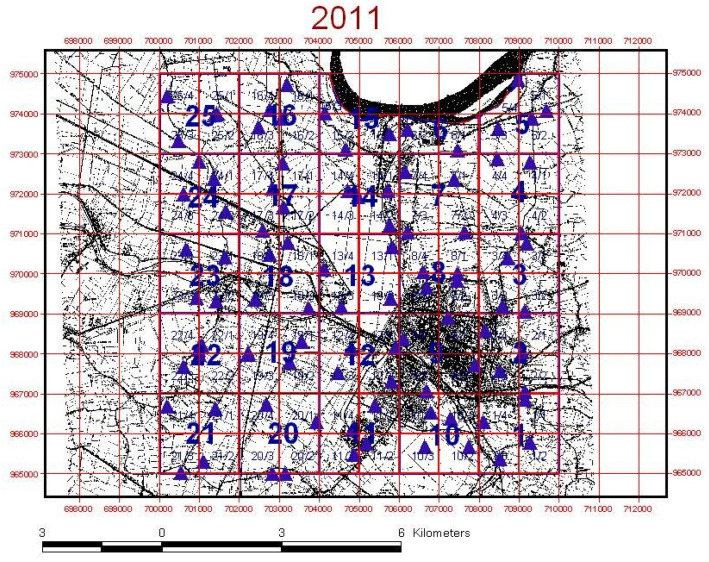
Moss-sampling sites in the municipality of Ferrara in 2011, from Gerdol et al. [24]. Clockwise and from the top, north, east, south and west are indicated. Blue triangles represent moss-sampling sites. Adapted with permission from [24], 2014, © Elsevier (Amsterdam, The Netherlands).

**Figure 2 ijerph-20-05591-f002:**
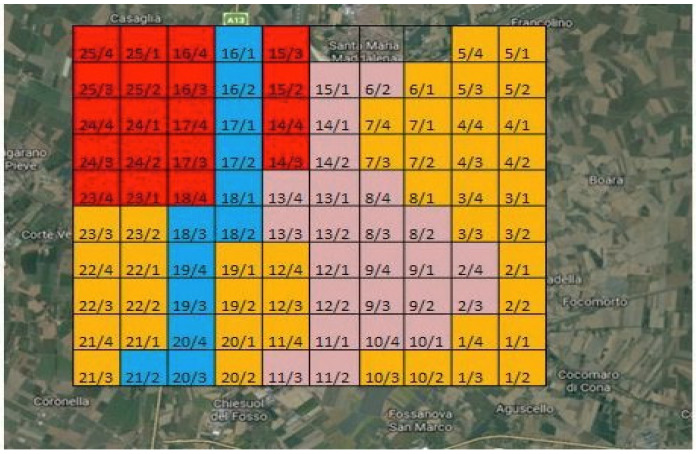
Four biomonitoring sampling sectors of the municipality of Ferrara according to environmental characteristics divided into sub-areas [24]: northwestern (NW, red), motorway (A13, blue), urban (Urb, pink) and rural (Rur, yellow) sectors.

**Figure 3 ijerph-20-05591-f003:**
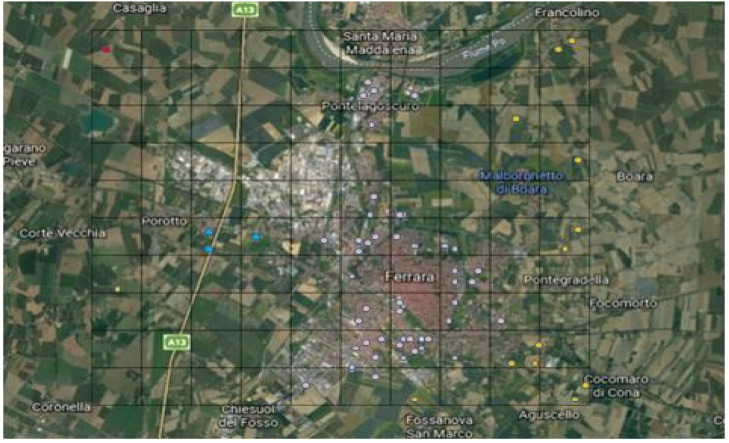
Mapping of ALS patients in subareas.

**Figure 4 ijerph-20-05591-f004:**
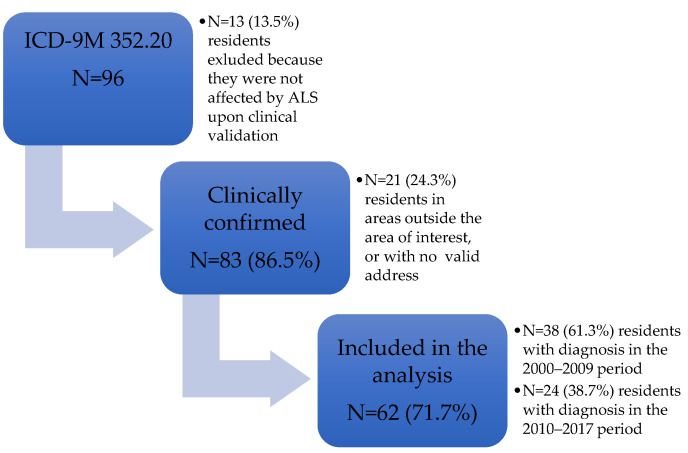
Flow chart of study participants.

**Table 1 ijerph-20-05591-t001:** Demographic characteristics of the study area’s general population, ALS cases and ALS incidence (per 100,000 inhabitants) in relation to urban vs. rural sectors, and to sex for the period 2000–2017.

	Urban (*N* = 54)	Rural (*N* = 13)	*p*
General population			
Total ^a^	1,587,273	326,638	
Men ^a^	734,995	151,251	
Women ^a^	852,278	175,386	
ALS cases: N (%)			
Total	46	13	
Men	25 (54.3)	6 (46.2)	ns
Women	21 (45.7)	7 (53.8)	ns
ALS incidence for the study period (per 100,000 inhabitants)			
Total	2.898	3.980	
Male	3.401	3.967	
Female	2.464	3.991	
Mean (SD) age at diagnosis (years)	71.1 (110.3)	69.4 (15.8)	ns

^a^: the overall estimated study population for the study period, 2000–2017. ns = not significant.

**Table 2 ijerph-20-05591-t002:** Correlation between ALS density and air pollutants in relation to study area, sectors of interest (urban and rural), sex and diagnosis period (2000–2009 and 2010–2017); Pearson coefficient (*ρ*), two-tailed significance; ns = not significant.

		Air Pollutants with Potential Neurodegenerative Role								
		Ag	Al	Cd	Cr	Cu	Fe	Mn	Pb	Se
Study area (all sectors)										
**Total**	** *ρ* **	0.104	−0.122	0.136	−0.059	0.758	−0.041	−0.098	0.073	−0.057
	*p*	ns	ns	ns	ns	0.000002	ns	ns	ns	ns
**Male**	** *ρ* **	−0.152	−0.237	−0.132	−0.186	0.608	−0.034	−0.212	−0.052	−0.131
	*p*	ns	ns	ns	ns	0.002	ns	ns	ns	ns
**Female**	** *ρ* **	0.056	−0.052	0.112	−0.012	0.782	0.028	0.067	−0.044	−0.126
	*p*	ns	ns	ns	ns	0.000028	ns	ns	ns	ns
**Urban sector**										
**Total**	** *ρ* **	−0.032	−0.196	0.056	−0.119	0.767	−0.096	−0.235	0.04	−0.161
	*p*	ns	ns	ns	ns	0.000128	ns	ns	ns	ns
**Male**	** *ρ* **	−0.374	−0.392	−0.197	−0.290	0.587	−0.105	−0.447	−0.088	−0.383
	*p*	ns	ns	ns	ns	0.017	ns	ns	ns	ns
**Female**	** *ρ* **	−0.066	−0.213	0.166	−0.065	0.872	−0.032	−0.058	−0.078	−0.195
	*p*	ns	ns	ns	ns	0.000047	ns	ns	ns	ns
**Rural sector**										
**Total**	** *ρ* **	0.11	−0.242	−0.249	−0.135	0.712	−0.127	0.065	0.377	−0.535
	*p*	ns	ns	ns	ns	0.048	ns	ns	ns	ns
**Male**		-	-	-	-	-	-	-	-	-
**Female**	** *ρ* **	0.447	0.08	−0.477	−0.177	0.667	−0.078	0.106	0.814	−0.394
	*p*	ns	ns	ns	ns	ns	ns	ns	ns	ns
**Cohort by diagnosis**										
**2000–2009**	** *ρ* **	0.051	−0.308	0.22	−0.242	0.724	−0.190	−0.159	0.49	0.039
	*p*	ns	ns	ns	ns	0.008	ns	ns	ns	ns
**2010–2017**	** *ρ* **	−0.135	−0.265	−0.056	−0.312	0.334	0.105	−0.095	0.377	0.211
	*p*	ns	ns	ns	ns	ns	ns	ns	ns	ns

**Table 3 ijerph-20-05591-t003:** Mean (±1 SE) levels of heavy metals detected in urban vs. rural sectors in years 2006 and 2011, respectively.

	Rural		Urban	
	2006	2011	2006	2011
Al	3152.88 ± 208.52	2668.27 ± 202.06	3119.42 ± 137.34	2824.71 ± 318.31
Ag	0.09 ± 0.03	0.04 ± 0.00	0.08 ± 0.01	0.08 ± 0.01
Cd	0.40 ± 0.03	0.42 ± 0.05	0.37 ± 0.02	0.55 ± 0.07
Cr	13.57 ± 1.02	14.06 ± 1.23	14.34 ± 1.73	19.77 ± 3.71
Cu	37.37 ± 4.77	57.96 ± 19.91	51.11 ± 19.14	40.43 ± 4.65
Fe (Yr ***)	5212.81 ± 359.67 a	3506.90 ± 240.13 b	4295.21 ± 165.33 ab	3818.83 ± 342.15 ab
Mn (Yr ***)	300.61 ± 16.21 a	153.81 ± 9.07 b	298.83 ± 16.36 a	166.33 ± 14.40 b
Pb	99.61 ± 57.18	44.13 ± 20.91	25.77 ± 2.44	67.37 ± 30.27
Se (Yr ***, A **, Yr × A *)	3.91 ± 0.45 b	0.52 ± 0.04 c	5.78 ± 0.64 a	0.68 ± 0.12 c

Mean (±1 SE) trace metal concentrations (μg g^−1^) resulting from two-way ANOVAs for year (Yr: 2006, 2011—df = 1.129) and area (A: rural and urban—df = 1.129) and their interaction (df = 2.129). Tukey’s post hoc test; *** *p* < 0.001, ** *p* < 0.01, and * *p* < 0.05. The means followed by the same letter within each line do not differ at *p* < 0.05.

## Data Availability

The dataset used for the analysis is available upon reasonable request to the corresponding author.

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
