# Peer review of "Amyotrophic Lateral Sclerosis and Air Pollutants in the Province of Ferrara, Northern Italy: An Ecological Study"

_ijerph, 2023, doi:10.3390/ijerph20085591_

Round 1

Reviewer 1 Report

 This manuscript describes an ecological study looking at the association between heavy metals in air pollution and amyotrophic lateral sclerosis (ALS)  in Ferrara, a region in Northern Italy. The authors report, in particular, that there was a correlation between  ALS density and copper concentration in (i) regions characterised as urban and rural and (ii) both men and women. The authors state that “our data supports the role of copper air pollutant to modulate the population susceptibility to neurodegeneration”.   This conclusion is, unfortunately, too definitive for two reasons. The first, as the authors clearly describe, is that this is an ecological study in which it is not possible to assign any exposure at an individual level and a person living in a “high” exposure area may not actually have  exposure higher than a person living In a “low” exposure area. Any associations observed in such studies are then, at best, potentially clues for further work to be undertaken.  The second reason is that their own data is not internally consistent ie  an association is observed in the 2000-2009 cohort but not with the 2010-2017 cohort. At best the data is , in part, consistent with a hypothesis linking copper pollution to ALS.

There are also some other issues that need to be clarified:

(i)       The authors need to indicate where the population data comes from

(ii)      the authors assume that the postcode at time of diagnosis is representative of where the person lived during the period of observation. Do the authors have any information as to whether the ALS cases lived at the same address during the time period? If not, can they described how this assumption affects their result.  

(iii)    line 114 “…crops release pollutants in this area”. How do crops do this?

(iv)    Figure 2. It is unclear how the sub-areas were coded into urban and rural sectors (in particular). For example areas 17/3 and 18/4 (in Figure 2)  are not coded as urban given that in Figure 3  they appear very similar to other areas eg 15/1 and 6/2 (in Figure 2) . Please provide more clearly, the reasons for coding these sectors.  

(v)      the description  (lines 153-158) of the calculations for the ALS case density indicate that the authors use the average annual population for the sector multiplied by the 17-year observation period. Is this correct for both the 2000-2009 and 2010-2017 cohorts as presumably the length of time should reflect the period of observation for these cohorts? Also by taking the total population, this will include people who are clearly not at risk for developing the disease during this time period (e.g. the new born). Did the authors also estimate associations in the population at risk for developing the disease? 

(vi)    The authors excluded a number of air pollutants (lines 159/160). Why did they not use them as negative controls in their analysis? Similarly, why did the authors not use another disease (with no known association with heavy metals) as a negative control for this study?

(vii)   The authors need to provide data on the levels of heavy metals detected in the samples analysed and particularly on the differences between urban and rural sectors so as to enable the reader to make a comparison to be made between these sectors.

Author Response

This manuscript describes an ecological study looking at the association between heavy metals in air pollution and amyotrophic lateral sclerosis (ALS) in Ferrara, a region in Northern Italy. The authors report, in particular, that there was a correlation between ALS density and copper concentration in (i) regions characterised as urban and rural and (i) both men and women. The authors state that "our data supports the role of copper air pollutant to modulate the population susceptibility to neurodegeneration". This conclusion is, unfortunately, too definitive for two reasons. The first, as the authors clearly describe, is that this is an ecological study in which it is not possible to assign any exposure at an individual level and a person living in a "high" exposure area may not actually have exposure higher than a person living In a "low" exposure area. Any associations observed in such studies are then, at best, potentially clues for further work to be undertaken. The second reason is that their own data is not internally consistent ie an association is observed in the 2000-2009 cohort but not with the 2010-2017 cohort. At best the data is, in part, consistent with a hypothesis linking copper pollution to ALS.

Thank you. We agree. We have accordingly diminished the strength of our statement in the Abstract.

There are also some other issues that need to be clarified:

  1. The authors need to indicate where the population data comes from

In the Materials and Methods section, specifically in sub-section 2.3. Ascertainment of ALS cases, the following integration was made:

“ALS cases were ascertained by using administrative data from the Ferrara University Hospital regional health information system. ICD-9M code 352.20 used for hospital admissions, including accesses to Day Hospital and Day Services for the period 2000-2017 were sought.

Personal code, sex, place and date of birth were then obtained, as well as place of residence on diagnosis, with post code. Diagnoses through ICD-9M code 352.20 were then validated against clinical records by neurologists M.P. and V.G. Exclusion criteria were i) misclassified diagnosis under ICD-9M code 352.20, ii) lack of information on place of residence on diagnosis, and iii) place of residence on diagnosis out of the study territory.”

2. the authors assume that the postcode at time of diagnosis is representative of where the person lived during the period of observation. Do the authors have any information as to whèther the ALS cases lived at the same address during the time period?

In the Materials and Methods section, specifically in sub-section 2.3. Ascertainment of ALS cases, we added a sentence to justify our assumption that the postcode at time of diagnosis is representative of where the person lived during the period of observation:

“Given the official administrative nature of the source of data, the patients’ residence addresses were up to date at the time of the study.

If not, can they described how this assumption affects their result.

                                    Not applicable.

3. line 114 ".. crops release pollutants in this area". How do crops do this?

In the Materials and Methods section, specifically in sub-section 2.5. Correlation matrices and study maps, the following explanation was given:

“Also, in this area pollutants are released in crops from intensive agriculture practice, such as through the use of agricultural machinery or widely dispersed pesticides and fertilizers.”

4. Figure 2. It is unclear how the sub-areas were code urban and rural sectors (in particular). For example area and 18/4 (in Figure 2) are not coded as urban given that in Figure 3 they appear very similar to other areas eg 15/1 and 6/2 (in Figure 2). Please provide more clearly, the reasons for coding these sectors.

About the sub-areas division, the squares were grouped into the four mentioned sectors, based on the distribution of CORINE land cover classes (reference: European Environment Agency, 2007. CLC2006 technical guidelines. CLC2006 technical guidelines — European Environment Agency (europa.eu).

In the Materials and Methods section, specifically in sub-section 2.5. Correlation matrices and study maps, we have revised accordingly:

“The squares were grouped into the following four sectors, based on the distribution of CORINE land cover classes [33].”

5. the description (lines 153-158) of the calculations for the ALS case density indicate that the authors use the average annual population for the sector multiplied by the 17-year observation period. Is this correct for both the 2000-2009 and 2010-2017 cohorts as presumably the length of time should reflect the period of observation for these cohorts?

ALS case density for the period 2000-2017 (17 years) was mapped for each sub-area in a specific sector and summed up for each sector. ALS case density was obtained dividing the number of cases for each sector and the reference population for that specific sector per 100,000 pop. The same estimate was obtained for the male and the female population, and separately for the diagnoses made in periods 2000-2009 and 2010-2017, respectively.

Also by taking the total population, this will include people who are clearly not at risk for developing the disease during this time period (e.g. the new born). Did the authors also estimate associations in the population at risk for developing the disease?

Consistently with literature on ALS epidemiology, the total population was considered as denominator.

6. The authors excluded a number of air pollutants (lines 159/160). Why did they not use them as negative controls in their analysis?

We thank the Reviewer for this comment. Other air pollutants were excluded from the analysis because given the ecologic (and not observational) nature of the study design, we ruled out pollutants with no know pathogenetic mechanism in neurodegeneration, so as to minimize ecological fallacy. The ecological study design does not imply use of controls.

Similarly, why did the authors not use another disease with no known association with heavy metals as a negative control for this study?

Similarly to what replied above (point 6) the ecological study design does not imply use of controls, neither among exposures nor among outcomes. Our sharp and specific interest was on ALS and considering another disease in the analysis, may have increase chances for ecological fallacy with the use of this specific study design.

7. The authors need to provide data on the levels of heavy metals detected in the samples analysed and particularly / the differences between urban and rural sectors so as to enable the reader to make a comparison to be made between these sectors.

A Table and a comment in the Results section, and a paragraph in the Discussion section was added about the levels of heavy metals detected in urban vs. rural sectors, respectively:

a. In the Results section: “The concentrations of the heavy metals included in the analyses, by year (2011 vs. 2011) and area (rural vs. urban) are shown in Table 3. A two-way ANOVA showed statistically significant differences in terms of years only for Fe and Mn, while for Se statistically significant differences were found both by year and by area, and by interaction between these two factors.”

Table 3. Mean (± 1 SE) levels of heavy metals detected in urban vs. rural sectors in years 2006 and 2011, respectively.

Rural

Urban

2006

2011

2006

2011

Al

3152.88 ± 208.52

2668.27 ± 202.06

3119.42 ± 137.34

2824.71 ± 318.31

Ag

0.09 ± 0.03

0.04 ± 0.00

0.08 ± 0.01

0.08 ± 0.01

Cd

0.40 ± 0.03

0.42 ± 0.05

0.37 ± 0.02

0.55 ± 0.07

Cr

13.57 ± 1.02

14.06 ± 1.23

14.34 ± 1.73

19.77 ± 3.71

Cu

37.37 ± 4.77

57.96 ± 19.91

51.11 ± 19.14

40.43 ± 4.65

Fe (Yr***)

5212.81 ± 359.67a

3506.90 ± 240.13 b

4295.21 ± 165.33 ab

3818.83 ± 342.15 ab

Mn (Yr***)

300.61 ± 16.21 a

153.81 ± 9.07 b

298.83 ± 16.36 a

166.33 ± 14.40 b

Pb

99.61 ± 57.18

44.13 ± 20.91

25.77 ± 2.44

67.37 ± 30.27

Se (Yr***, A**, Yr×A*)

3.91 ± 0.45 b

0.52 ± 0.04 c

5.78 ± 0.64 a

0.68 ± 0.12 c

Mean (± 1 SE) trace metal concentrations (μg g-1 ) resulting from two-way ANOVAs for year (Yr: 2006, 2011 - df = 1,129), area (A: rural, urban - df = 1,129) and their interaction (df = 2,129). Tukey’s post-hoc *** p<0.001, **p<0.01, *p<0.05. The means followed by the same letter within each line do not differ at p<0.05.

b. Discussion section: Of note, the pathogenesis of ALS is multifactorial and depends on a complex (and only partially understood) combination of genetic, environmental and metabolic factors [19, 72-75]. It is therefore possible to speculate that Cu acts synergistically with other risk factors and that only after exceeding a certain critical threshold does it contribute to the risk of ALS onset [75].

Reviewer 2 Report

The study by Antonioni et al. is concerned with analyzing the existence of a possible correlation between exposure to metals in different areas of the province of Ferrara (northern Italy) and the incidence of cases of amyotrophic lateral sclerosis, during the time period between 2007 and 2017. The study is outlined with an ecological design, and it is proposed to use the concentration of metals detected in lichens and mosses in the geographical reference area as a biomonitoring parameter.

Although numerous evidences have supported the validity of the methodology used by the authors as a biomonitoring tool, and although both the introduction and the discussion of the manuscript are well structured and report numerous references inherent to the topic, nevertheless the overall design of the study shows numerous critical issues. 

First, in assessing the association between environmental risk factors and disease incidence, an ecological study design is not entirely appropriate in the case of rare diseases with multifactorial etiology, whereas it is usually used in the case of common and widespread diseases in the population. 

Second, the authors refer exclusively to cases reported by a single hospital in Ferrara, neglecting to consider any diagnoses reported in other hospitals, for example, of a private nature, in the area.

In addition, numerous confounding factors, such as work activity, the presence of the same indoor pollutants, and other concomitant exposures cannot be assessed with this type of study, and make the results obtained exclusively speculative and unsupported by the methodologies used.

Finally, the statistical analyses used are not appropriate to the study design chosen by the authors, which usually makes use of generalized additive models (GAMs) to study the association between disease incidence and exposure to different environmental factors.

Author Response

The study by Antonioni et al. is concerned with analyzing the existence of a possible correlation between exposure to metals in different areas of the province of Ferrara (northern Italy) and the incidence of cases of amyotrophic lateral sclerosis, during the time period between 2007 and 2017. The study is outlined with an ecological design, and it is proposed to use the concentration of metals detected in lichens and mosses in the geographical reference area as a biomonitoring parameter. Although numerous evidences have supported the validity of the methodology used by the authors as a biomonitoring tool, and although both the introduction and the discussion of the manuscript are well structured and report numerous references inherent to the topic, nevertheless the overall design of the study shows numerous critical issues.

First, in assessing the association between environmental risk factors and disease incidence, an ecological study design is not entirely appropriate in the case of rare diseases with multifactorial etiology, whereas it is usually used in the case of common and widespread diseases in the population.

We thank the Reviewer for his/her valuable comments.

Despite being ecological study designs most suitable for common diseases and widespread throughout the population, this methodological strategy allows to correlate maps of exposures and outcomes deriving from large populations, possibly overcoming the complex and rare multifactorial diseases. Here are some example:

Alanazi AFR, Naser AY, Pakan P, Alanazi AF, Alanazi AAA, Alsairafi ZK, Alsaleh FM. Trends of Hospital Admissions Due to Congenital Anomalies in England and Wales between 1999 and 2019: An Ecological Study. International Journal of Environmental Research and Public Health. 2021; 18(22):11808. https://doi.org/10.3390/ijerph182211808

Reynolds P, Von Behren J, Gunier RB, Goldberg DE, Hertz A, Harnly ME. Childhood cancer and agricultural pesticide use: an ecologic study in California. Environ Health Perspect. 2002 Mar;110(3):319-24. doi: 10.1289/ehp.02110319. PMID: 11882484; PMCID: PMC1240773.

Kamwendo F, Forslin L, Bodin L, Danielsson D. Epidemiology of ectopic pregnancy during a 28 year period and the role of pelvic inflammatory disease. Sex Transm Infect. 2000 Feb;76(1):28-32. doi: 10.1136/sti.76.1.28. PMID: 10817065; PMCID: PMC1760576.

To better clarify this, we have added a specific sentence about the limitations of our study (Discussion section):

“Moreover, although ecological study designs are more suitable for common and widespread diseases, several examples of such methodology used to investigate rare diseases and diseases with multifactorial aetiology are reported in the literature [e.g. 78-80].”

Second, the authors refer exclusively to cases reported by a single hospital in Ferrara, neglecting to consider any diagnoses reported in other hospitals, for example, of a private nature, in the area.

Indeed the Ferrara Hospital IT service provided us with administrative data which derive from regional health information system and we were thus able to collect all cases with ICD9-CM code for ALS residing in the province of Ferrara at diagnosis and in the period 2000-2017. Because the Ferrara University Hospital is a hub centre for rare diseases (ALS), basically all these patients received a diagnosis at this Hospital.

We have already clarified this in the Material and Methods section, sub-section 2.3. Ascertainment of ALS cases.

In addition, numerous confounding factors, such as work activity, the presence of the same indoor pollutants, and other concomitant exposures cannot be assessed with this type of study, and make the results obtained exclusively speculative and unsupported by the methodologies used.

Thank you for this comment. Some studies have shown that, although indoor pollutants may contribute to the risk of disease, outdoor pollutants are more relevant, especially in adults. For example, please see:

Zhou L, Liu G, Shen M, Liu Y, Lam PKS. Characteristics of indoor dust in an industrial city: Comparison with outdoor dust and atmospheric particulates. Chemosphere. 2021 Jun;272:129952. doi: 10.1016/j.chemosphere.2021.129952. Epub 2021 Feb 12. PMID: 33601210.

Cao S, Wen D, Chen X, Duan X, Zhang L, Wang B, Qin N, Wei F. Source identification of pollution and health risks to metals in household indoor and outdoor dust: A cross-sectional study in a typical mining town, China. Environ Pollut. 2022 Jan 15;293:118551. doi: 10.1016/j.envpol.2021.118551. Epub 2021 Nov 20. PMID: 34813887.

Furthermore, the study design involved correlating the ALS distribution map with that of air pollutants by assessing their levels in mosses and lichens. This certainly prevented other important factors from being considered, but these limitations are inherent to the study strategy used.

Also, considering that, especially in rural areas, people spend more time outdoors (for example, please see: Kelly P and Lobao L. The Social Bases of Rural-Urban Political Divides: Social Status, Work, and Sociocultural Beliefs. Rural Sociology 2019, 84: 669-705. https://doi.org/10.1111/ruso.12256), this supported our decision to focus on outdoor pollutants only in this study.

We have integrated the Discussion section:

“In our study we privileged correlating ALS concentration with outdoor rather than indoor pollutants given their potential higher impact in the disease pathogenesis as shown elsewhere [81-83].”

Finally, the statistical analyses used are not appropriate to the study design chosen by the authors, which usually makes use of generalized additive models (GAMs) to study the association between disease incidence and exposure to different environmental factors.

Ecological studies rely on statistical correlations and we then acknowledge less rigorous statistical outputs than association studies. Nevertheless, they may disclose interesting patterns of relationships between maps of exposures and outcomes, to eventually explore through association studies. Based on Kain MP, Bolker BM, McCoy MW. 2015. A practical guide and power analysis for GLMMs: detecting among treatment variation in random effects. PeerJ 3:e1226 https://doi.org/10.7717/peerj.1226, we also thought that the generalized additive models (GAMs) was less suitable for our study design, considering the relatively small sample size. Our decision was also endorsed by a biostatistician experienced in neuroepidemiological studies.

Reviewer 3 Report

The manuscript prepared by the interdisciplinary group of scientists from University of Ferrara is original, well written, concise in form and presents important public health problem : long-term analysis of incidence of ALS in the province of Ferrara and relationship with several environmental air pollutants, especially heavy metals. 

I do not have any critical remarks related to the materials and methods as well as conclusions of the study. The authors described properly several limitations of their own study.

In my opinion it would be desirable to complete the discussion with some epidemiological data presenting the incidence of ALS and mortality trends in selected countries in the world and to compare the data available in literature  with their own study. There exist good papers from Spain, New Zealand and United States ( f.e Santurtun et al 2016, Cao et al 2018 or Schwartz and Klug 2016).

Author Response

The manuscript prepared by the interdisciplinary group of scientists from University of Ferrara is original, well written, concise in form and presents important public health problem: long-term analysis of incidence of ALS in the province of Ferrara and relationship with several environmental air pollutants, especially heavy metals.

I do not have any critical remarks related to the materials and methods as well as conclusions of the study. The authors described properly several limitations of their own study. In my opinion it would be desirable to complete the discussion with some epidemiological data presenting the incidence of ALS and mortality trends in selected countries in the world and to compare the data available in literature with their own study. There exist good papers from Spain, New Zealand and United States (fe Santurtun at al 2016, Cao et al 2018 or Schwartz and Klug 2016).

We are grateful for your appreciation of our work and for your valuable suggestions, which will certainly improve the quality of our paper. We have added in the Discussion section a paragraph on data and comparison with other countries as you suggested. In addition, the bibliography has been implemented with additional references:

“ALS incidence in the province of Ferrara is consistent with that of other European countries, i.e., 1-3/100,000 pop./year [23, 64, 65]. Santurtún et al have shown, in a Spanish population, a clear positive relationship between ALS and air Pb levels [65], pointing to more than one air heavy metal pollutant in association with ALS increased risk [66, 67]. Stable ALS mortality and hence incidence rates over the past decades have also been reported overseas. A recent study in New Zealand showed a mortality rate of 2.8/100,000 pop./year in the period 1992-2013 with 74% of the New Zealand population made up of European and Caucasian descendants [68]. In the US population very similar incidence and mortality rates are reported with the exception of Midwest where the registered mortality from ALS is significantly higher [69-71].”